# Index Matrix-Based Modeling and Simulation of Buck Converter

**Nikolay Hinov** [1,*] , **Polya Gocheva** [2] **and Valeri Gochev** [3]

1    Department of Power Electronics, Technical University of Sofia, 1756 Sofia, Bulgaria
2    Doctoral School, Technical University of Sofia, 1756 Sofia, Bulgaria; p.gocheva@utp.bg
3    Department of Telecommunications, University of Telecommunications and Posts, 1700 Sofia, Bulgaria;
     v.gochev@utp.bg
*    Correspondence: hinov@tu-sofia.bg

**Abstract:** The approach described in this paper handles the parameters and characteristics (analog and discrete ones) of a Buck DC-DC converter (in its power and control parts) in a common manner. The usage of probably complicated differential equations for discrete dynamical systems is avoided by means of index matrix equations, which can be easily understood. Compared to classical matrix models, the proposed index matrix models are more descriptive and also smaller in size. Such functionality is widely applied by the authors, and a new operation is defined and used as well. The relevance of the proposed techniques in power electronics, because of switching topologies and a limited numbers of components, is argued. Respective examples of functioning modes of the considered converter, in which power circuits and controllers are modeled jointly, are given. Estimations of analog values are based on partly linear dependencies, which are shown to be adequate in first-order analyses. Specific expressions for a Buck DC-DC converter are presented. A model-solving technique and an exhaustive search on a parameter space are considered in detail and automated via well-formalized algorithms. Nested parameter intervals and verifications with normal probability distributions are used in an optimization procedure. The full agreement of implementation via MATLAB source code with results obtained via Simulink is demonstrated. The short simulation times of this software (compared to Simulink and a .NET desktop application developed by the authors) justify the search for optimal variants in a wide multi-dimensional space. A max–min procedure with 10,000 simulations in each verification step is presented.

**Keywords:** index matrix; model-based design; optimization; parameter space; power electronics

**MSC:** 03C98; 11C20

## 1. Introduction

Due to the growing need to convert electricity with electronic means, power electronics is a rapidly developing field. This fact requires new developments and an improvement in the tools used to obtain the optimal design of power electronic devices and systems. On the other hand, because of the presence of power and control parts (controllers), which are quite different and mutually influence and complement each other, in many approaches, electronic converters are not susceptible to modeling as a whole unified system. One of the main purposes of the authors' papers has been to present a joint technique based on the apparatus of index matrices [1,2].

First-order time-invariant dynamical systems are often used as modeling tools [3,4]. Discrete dynamical systems are represented in [5] via equations with constant classical matrices, and this approach does not differ significantly from well-known control system engineering methods, in which derivatives of state variables are written as linear combinations of system states and inputs [4,6]. Both of the aforementioned techniques determine a simple stability analysis of discrete or analog values in multi-input–multi-output systems, including in power electronic devices [7,8].

In [9], two ideal switches, which work synchronously in a Boost DC-DC converter, determine two functioning modes, and, therefore, two different constant matrices in a process of discrete dynamical modeling. In the present paper, each switch is considered to have two modes too, but the transistor and the diode work out of sync. Such a principle requires variants to be checked and, similarly, determines the exponential dependence of the number of variants on the number of switches [1]. However, the latter is not large in power electronic devices; in the following sections, the transistor and the diode determine four modes. Generally, the switching topologies in power converters do not allow an analysis based on constant matrices or constant differential equations. Therefore, computer simulations are widely used; for example, the graphical programming environment Simulink, which is shown in the present paper in a comparison, provides a set of solvers and has power electronic tools [10].

A feature that dynamical systems have in common is that they are all simulated by estimating states at consecutive model time moments. The obtainment of values on system states (which usually are presented by one-dimensional or multi-dimensional arrays) is known as model solving. No single solving method is optimal for all systems. For example, a typical power conversion system is constituted by subsystems, which have very different levels of detail [11,12]. However, in the present paper, index matrix equations are used in the configuration of electronic components. The discretization step in the algorithm proposed in Section 4 is chosen to be constant; it is obvious that decreasing it increases the accuracy and vice versa (similar dependences are demonstrated in a figure in Section 4). Values on system states are estimated at regular model time intervals from the beginning to the end of the simulation; i.e., a fixed-step solver is constructed. Moreover, the index matrix describing system states can vary in size according to a new kind of multiplication, which is defined for the first time in the present article.

Index matrices (this concept was introduced in 1984) extend classical matrix functionality; their properties have been studied and summarized in [13,14]; basic notes are given in Section 2, too. The index matrices extend the concept of the classical matrices and provide clear correspondence between their elements and indices. Different types of their elements have been proposed; in the present paper, real numbers are used. The index matrices are mainly connected to generalized nets, in which index matrices are basic components [15,16]. Applications of index matrices concern mathematical and real-life objects, for example, databases, but the usage of this apparatus in the modeling of electrical circuits is very rare. A full review of their applications is unpublished as of yet. Indeed, for a long time, they have been used with Boolean values and integers only as an auxiliary tool in generalized net transitions.

Data structures, which implement index matrices in the programming class *IndexMatrix*, are demonstrated in [1]. There are implementations of all operations from Section 2, except the new one, in this class too. The .NET application, which is presented in [1], concerns analog and digital electronic components and circuits, model solvers, graphics, etc. Its basic feature is the shorter simulation duration compared to Simulink; it has about a 20 times shorter runtime in the case of the considered buck converter and about a 10 times shorter runtime in the case of a three-phase converter. Generally, all model-based design techniques require many simulations [17,18]; the optimization algorithm from Section 5 uses a large number of simulations in a verification process.

The approach discussed In this paper is implemented via MATLAB source code [19]. This choice is natural because classical matrices are presented in the algorithm from Section 4. The core functionality of MATLAB is closely related to matrix manipulation. The obtained runtimes ensure the model's ability to perform exhaustive searches; this fact is a prerequisite for the application of artificial intelligence techniques, too [2]. The simulation durations of this implementation, as well as those in the cases of the discussed .NET application and Simulink, are given in Section 6.

## 2. Basic Definitions of Index Matrices with Real Number Elements

All index matrices with real numbers as elements (R-IM) [13,14] have the form

$$\overline{A} = \left[ K, \quad L, \quad \{a_{k,l}\}_{k \in K, l \in L} \subset \mathbb{R} \right] \tag{1}$$

with index sets (called the first and second one, respectively)

$$K = \{k_1, \quad k_2, \quad \ldots, \quad k_m\}, \qquad L = \{l_1, \quad l_2, \quad \ldots, \quad l_n\}. \tag{2}$$

In the present paper (as well as in [1]), the following notations of K and L are used:

$$K = I_1(\overline{A}), \qquad L = I_2(\overline{A}). \tag{3}$$

The usual graphical representation of the above index matrix form is

$$\overline{A} = \begin{array}{c|cccc} & l_1 & l_2 & \cdots & l_n \\ \hline k_1 & a_{k_1,l_1} & a_{k_1,l_2} & \cdots & a_{k_1,l_n} \\ k_2 & a_{k_2,l_1} & a_{k_2,l_2} & \cdots & a_{k_2,l_n} \\ \vdots & \vdots & \vdots & \ddots & \vdots \\ k_m & a_{k_m,l_1} & a_{k_m,l_2} & \cdots & a_{k_m,l_n} \end{array} \ . \tag{4}$$

Lines divide indices from elements. All indices from the index set are used like lower indices, too. Each element is positioned in the same row with the index which is identical to its first lower index and in the same column with the index which is identical to its second lower index. The ordering of the indexes explicitly determines the ordering of the elements. Swaps of rows or columns (along with the indexes) of an index matrix do not change it. For comparison, in the classical matrices, row and column indexes are implicitly used, and any swap of elements must generally correspond to respective swaps in other objects in order dependencies (for example, in matrix equations) to be preserved.

Authors' notations of index and classical matrices are proposed here; they lead to clear expressions, those in algorithms (1), (2) and respectively (4) can be represented by a classical matrix (which is denoted by "CM" in the present paper)

$$A' = \mathrm{CM}_{U,V} \, \overline{A} = \begin{pmatrix} a_{u_1,v_1} & a_{u_1,v_2} & \cdots & a_{u_1,v_n} \\ a_{u_2,v_1} & a_{u_2,v_2} & \cdots & a_{u_2,v_n} \\ \vdots & \vdots & \ddots & \vdots \\ a_{u_m,v_1} & a_{u_m,v_2} & \cdots & a_{u_m,v_n} \end{pmatrix} \tag{5}$$

and ordered sets

$$U = \langle u_1, u_2, \ldots, u_m \rangle, \qquad V = \langle v_1, v_2, \ldots, v_n \rangle, \tag{6}$$

where U and V are permutations of $I_1(\overline{A})$ and $I_2(\overline{A})$, respectively. There are m!n! options for (5) according to all permutations in (6). It is important to note that these variants differ only by ordering. The inverse notation

$$\mathrm{IM}_{U,V} A' = \overline{A} \tag{7}$$

on $\mathrm{CM}_{U,V} \, \overline{A} = A'$ is also used in the present paper.

Brief notes of basic operations (summation, multiplication, transposing and projecting of index matrices—see [12,13]) are given below in order of specific usages (for example, in summation with non-intersecting index sets) and a new definition of a multiplication to be argued.

An example of a transpose R-IM is

$$\overline{G} = \overline{A}^T = \left[ K, \quad L, \quad \{a_{k,l}\}_{k \in K, l \in L} \right]^T = \left[ L, \quad K, \quad \{g_{l,k} \equiv a_{k,l}\}_{l \in L, k \in K} \right]. \tag{8}$$

Each projection of $\overline{A}$ (this one is an index sub-matrix, denoted by $\overline{H}$ below, with notation $\overline{H} \subseteq \overline{A}$ in the present paper) has the form

$$\overline{H} \equiv \text{pro}_{R,S}\overline{A} = \begin{bmatrix} R, & S, & \{a_{r,s}\}_{r\in R, s\in S} \end{bmatrix} \subseteq \overline{A}, \qquad R\subseteq K, \qquad S\subseteq L. \tag{9}$$

Let R-IM

$$\overline{B} = \begin{bmatrix} P, & Q, & \{b_{p,q}\}_{p\in P, q\in Q} \end{bmatrix} \tag{10}$$

be given. Sum

$$\overline{A} \oplus \overline{B} = \begin{bmatrix} K\cup P, & L\cup Q, & \{c_{r,s}\}_{r\in K\cup P, s\in L\cup Q} \end{bmatrix},$$

$$c_{r,s} = \begin{cases} a_{r,s}, & \text{if } (r\in K \wedge s\in L-Q)\vee(r\in K-P \wedge s\in L); \\ b_{r,s}, & \text{if } (r\in P \wedge s\in Q-L)\vee(r\in P-K \wedge s\in Q); \\ a_{r,s} + b_{r,s}, & \text{if } (r\in K\cap P \wedge s\in L\cap Q;) \\ 0, & \text{otherwise}; \end{cases} \tag{11}$$

is a R-IM. Formally, all classical matrix sums can be represented by a proper case of the upper R-IM sum, for which $K \equiv P$ and $L \equiv Q$ (see value $a_{r,s} + b_{r,s}$ in the third case of $c_{r,s}$). Oppositely, if there is a given lack of intersection of index sets, this value can be excluded from (11):

$$(K\cap P = \varnothing \vee L\cap Q = \varnothing) \Rightarrow c_{r,s} = \begin{cases} a_{r,s}, & \text{if } r\in K \wedge s\in L; \\ b_{r,s}, & \text{if } r\in P \wedge s\in Q; \\ 0, & \text{otherwise}; \end{cases} \tag{12}$$

Here, all elements of $\overline{A}$ and $\overline{B}$ are presented in different parts of $\overline{A} \oplus \overline{B}$.

The product of $\overline{A}$ and $\overline{B}$

$$\overline{A} \odot \overline{B} = \begin{bmatrix} K\cup(P-L), & Q\cup(L-P), & \{d_{r,s}\}_{r\in K\cup(P-L), s\in Q\cup(L-P)} \end{bmatrix},$$

$$d_{r,s} = \begin{cases} a_{r,s}, & \text{if } r \in K \wedge s \in L-P-Q; \\ b_{r,s}, & \text{if } r \in P-K-L \wedge s \in Q; \\ \sum\limits_{t\in L\cap P} a_{r,t}b_{t,s}, & \text{if } r \in K \wedge s \in Q; \\ 0, & \text{otherwise}; \end{cases} \tag{13}$$

is an R-IM, too. Formally, all classical matrix products can be represented by a proper case of the upper R-IM product, for which $L \equiv P$ (see the sum in its common form).

A new kind of an R-IM product is defined below by the authors:

$$\overline{A} \otimes \overline{B} = \begin{bmatrix} K, & Q\cup(L-P), & \{d'_{r,s}\}_{r\in K, s\in Q\cup(L-P)} \end{bmatrix} \tag{14}$$

$$d'_{r,s} = \begin{cases} a_{r,s}, & \text{if } s \in L-P-Q; \\ \sum\limits_{t\in L\cap P} a_{r,t}b_{t,s}, & \text{if } s \in Q. \end{cases} \tag{15}$$

It reduces $\overline{A} \odot \overline{B}$ according to its rows (the first index sets of $\overline{A}$ and $\overline{A} \otimes \overline{B}$ are identical):

$$\overline{A} \otimes \overline{B} = \text{pro}_{I_1(\overline{A}), I_2(\overline{A} \odot \overline{B})}(\overline{A} \odot \overline{B}). \tag{16}$$

The following statements hold:

$$L\subseteq P \Rightarrow \overline{A} \odot \overline{B} = \begin{bmatrix} K\cup(P-L), & Q, & \{d''_{r,s}\}_{r\in K\cup(P-L), s\in Q} \end{bmatrix},$$

$$d''_{r,s} = \begin{cases} b_{r,s}, & \text{if } r \in P-K-L; \\ \sum\limits_{t\in L} a_{r,t}b_{t,s}, & \text{if } r \in K. \end{cases} \tag{17}$$

$$L \subseteq P \quad \Rightarrow \quad \overline{A} \otimes \overline{B} = \left[ K, \quad Q, \quad \{d'''_{r,s}\}_{r\in K, s\in Q} \right], \qquad d'''_{r,s} = \sum\limits_{t\in L} a_{r,t}b_{t,s}. \tag{18}$$

Examples of the new R-IM multiplication are given in the next sections.

## 3. R-IM Models in Electronic Circuit Design

### 3.1. Models of Electronic Components

Models of basic electronic components are given below. Similar but simpler ones are described in [1], in which ideal components are used; additional resistances and potential barriers are taken into account below.

Let time set $\Theta$ with sampling step $\Delta t$ and discretization frequency $F_d$ be given under the following restrictions:

$$\Theta \equiv \{t_0, \quad t_0 + \Delta t, \quad t_0 + 2\Delta t, \quad \ldots\}, \quad t_0, \Delta t \in \mathbb{R}, \quad \Delta t = \frac{1}{F_d} > 0.$$

An R-IM, which concerns an inductor with inductance L and additional resistance $R_L$, is presented at first:

$$\overline{F_L} = \begin{array}{c|ccc} & u_L^{cu} & i_L^{cu} & i_L^{pr} \\ \hline c_L & -1 & \frac{L}{\Delta t} + R_L & -\frac{L}{\Delta t} \end{array}. \tag{19}$$

$c_L$ denotes the inductor. $u_L^{cu}$, $i_L^{cu}$, $u_L^{pr}$ and $i_L^{pr}$ indicate voltages (with "u") and amperages (with "i") in current ($t \in \Theta$, marked by "cu") and previous ($t - \Delta t$, marked by "pr") time moments. This notation is used by the authors in [1] too; index matrix

$$\overline{X_L}(t) = \left( \begin{array}{c|cccc} & u_L^{cu} & i_L^{cu} & u_L^{pr} & i_L^{pr} \\ \hline o & u_L(t) & i_L(t) & u_L(t-\Delta t) & i_L(t-\Delta t) \end{array} \right)^T, \tag{20}$$

where o is a common index, contains all these four indexes. The R-IM equation

$$\overline{F_L} \otimes \left[ \overline{X_L}(t) \oplus \begin{array}{c|c} & o \\ \hline o & 1 \end{array} \right] \equiv$$

$$\equiv \begin{array}{c|c} & o \\ \hline c_L & -u_L(t) + \left( \frac{L}{\Delta t} + R_L \right) i_L(t) - \frac{L}{\Delta t} i_L(t-\Delta t) \end{array} = \overline{O} \tag{21}$$

($\overline{O}$ is a zero R-IM—its only element is equal to zero) determines equality

$$u_L(t) = L \frac{i_L(t) - i_L(t-\Delta t)}{\Delta t} + R_L i_L(t), \tag{22}$$

which is an approximation of differential equation

$$u_L(t) = L \frac{di_L(t)}{dt} + R_L i_L(t). \tag{23}$$

Equation (21) is based on the new R-IM multiplication (14). For comparison,

$$\overline{F_L} \odot \left[ \overline{X_L}(t) \oplus \begin{array}{c|c} & o \\ \hline o & 1 \end{array} \right] \equiv \begin{array}{c|c} & o \\ \hline c_L & -u_L(t) + \left( \frac{L}{\Delta t} + R_L \right) i_L(t) - \frac{L}{\Delta t} i_L(t-\Delta t) \\ u_L^{pr} & u_L(t-\Delta t) \\ o & 1 \end{array} \tag{24}$$

Similarly, in case of a capacitor with capacitance C and additional resistance $R_C$ approximation

$$i_C(t) = C \frac{[u_C(t) - R_C i_C(t)] - [u_C(t-\Delta t) - R_C i_C(t-\Delta t)]}{\Delta t} \tag{25}$$

can be presented by

$$\overline{F_C} = \begin{array}{c|cccc} & u_C^{cu} & i_C^{cu} & u_C^{pr} & i_C^{pr} \\ \hline c_C & \frac{C}{\Delta t} & -1 - \frac{CR_C}{\Delta t} & -\frac{C}{\Delta t} & \frac{CR_C}{\Delta t} \end{array}. \tag{26}$$

In the same manner, index matrix $\overline{X_C}(t)$ with four elements (see (20)) and an analog of Equation (21) can be formed.

The presented approach can be extended to other power components from Figure 1, and common index matrices can be constructed:

$$\overline{F_{PCs}}(t) = \overline{F_E} \oplus \overline{F_{T_{out}}}(t) \oplus \overline{F_D}(t) \oplus \overline{F_L} \oplus \overline{F_C} \oplus \overline{F_R}; \tag{27}$$

$$\overline{X_{PCs}}(t) = \overline{X_E}(t) \oplus \overline{X_{T_{out}}}(t) \oplus \overline{X_D}(t) \oplus \overline{X_L}(t) \oplus \overline{X_C}(t) \oplus \overline{X_R}(t). \tag{28}$$

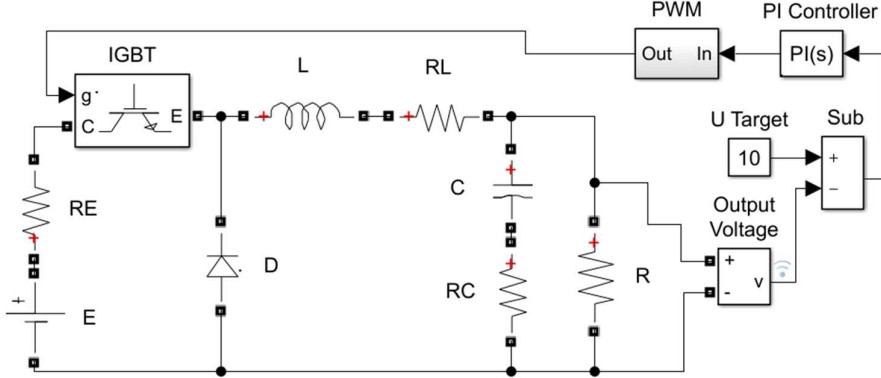

**Figure 1.** Simulink model of buck DC-DC converter with PI controller.

Abbreviation "PCs" means "power components" (all of the last ones, which are presented below, are two-terminal except the transistor). $\overline{F_{T_{out}}}(t)$ and $\overline{X_{T_{out}}}(t)$ concern the transistor in its emitter–collector junction.

The next R-IM models the diode functioning:

$$\overline{F_D}(t) = \begin{cases} \overline{F_D^1} \equiv \begin{array}{c|ccc} & u_D^{cu} & i_D^{cu} & o \\ \hline c_D & -1 & R_D & U_D^0 \end{array}, & \text{if } u_D(t) > U_D^0; \\[2ex] \overline{F_D^0} \equiv \begin{array}{c|c} & i_D^{cu} \\ \hline c_D & 1 \end{array}, & \text{otherwise.} \end{cases} \tag{29}$$

It leads to the following statements:

$$u_D(t) > U_D^0 \Rightarrow u_D(t) = U_D^0 + R_D i_D(t). \tag{30}$$

$$u_D(t) \leq U_D^0 \Rightarrow i_D(t) = 0. \tag{31}$$

Here, $U_D^0$ and $R_D$ denote a threshold voltage (a potential barrier) and a dynamic resistance, respectively.



The next R-IM models the transistor emitter–collector junction functioning:

$$\overline{F_{T_{out}}}(t) = \begin{cases} \overline{F_{T_{out}}^1} \equiv \dfrac{\begin{array}{|ccc}u_{T_{out}}^{cu} & i_{T_{out}}^{cu} & o \\ \hline -1 & R_{T_{out}} & U_{T_{out}}^0\end{array}}{c_{T_{out}}}, & \text{if } s_{T_{in}}(t) = 1 \wedge u_{T_{out}}(t) > U_{T_{out}}^0; \\[4mm] \overline{F_{T_{out}}^0} \equiv \dfrac{\begin{array}{|c}i_{T_{out}}^{cu} \\ \hline 1\end{array}}{c_{T_{out}}}, & \text{otherwise.} \end{cases} \tag{32}$$

Similarly to the diode model, it determines the following statements:

$$s_{T_{in}}(t) = 1 \wedge u_{T_{out}}(t) > U_{T_{out}}^0 \Rightarrow u_{T_{out}}(t) = U_{T_{out}}^0 + R_{T_{out}} i_{T_{out}}(t), \tag{33}$$

$$s_{T_{in}}(t) = 0 \vee u_{T_{out}}(t) \le U_{T_{out}}^0 \Rightarrow i_{T_{out}}(t) = 0; \tag{34}$$

with threshold voltage $U_{T_{out}}^0$ and dynamic resistance $R_{T_{out}}$; signal $s_{T_{in}}(t)$ on the input of the IGBT is digital.

$\overline{F_R}$ and $\overline{F_E}$, which model the resistor and the voltage source, are simpler:

$$\overline{F_R} = \dfrac{\begin{array}{|cc}u_R^{cu} & i_R^{cu} \\ \hline -1 & R\end{array}}{c_R}; \qquad \overline{F_E} = \dfrac{\begin{array}{|ccc}u_E^{cu} & i_E^{cu} & o \\ \hline -1 & R_E & E\end{array}}{c_E}. \tag{35}$$

The following equations (see (27) and (28)) can be used with the upper R-IMs:

$$\overline{F_{PCs}} \otimes \left[ \overline{X_{PCs}}(t) \oplus \begin{array}{|c}o \\ \hline 1\end{array} \right] = \overline{O}; \tag{36}$$

$$\overline{F_{PC}} \otimes \left[ \overline{X_{PC}}(t) \oplus \begin{array}{|c}o \\ \hline 1\end{array} \right] = \overline{O}, \qquad PC \in \{E, \ T_{out}, \ D, \ L, \ C, \ R\}. \tag{37}$$

The components of the considered control circuit (see Figure 1) can be modeled by R-IMs in a similar manner [1]. The subtraction block with target voltage $U_{Target}$, the proportional–integral (PI) controller and the pulse-width modulation (PWM) block are presented by $\overline{F_{Sub}}$, $\overline{F_{PI}}$ and $\overline{F_{PWM}}(t)$, respectively, below:

$$\overline{F_{Sub}} = \dfrac{\begin{array}{|ccc}u_{Sub_{in}}^{cu} & u_{Sub_{out}}^{cu} & o \\ \hline -1 & -1 & U_{Target}\end{array}}{c_{Sub}}; \tag{38}$$

$$\overline{F_{PI}} = \dfrac{\begin{array}{|cccc}u_{PI_{out}}^{cu} & u_{PI_{out}}^{pr} & u_{PI_{in}}^{cu} & u_{PI_{in}}^{pr} \\ \hline -\frac{1}{\Delta t} & \frac{1}{\Delta t} & \frac{K_p}{\Delta t} + K_i & -\frac{K_p}{\Delta t}\end{array}}{c_{PI}}; \tag{39}$$

$$\overline{F_{PWM}}(t) = \begin{cases} \dfrac{\begin{array}{|cc}s_{PWM_{out}}^{cu} & o \\ \hline -1 & 1\end{array}}{c_{PWM}}, & \text{if } u_{Triang}(t) < u_{PWM_{in}}(\psi); \\[4mm] \dfrac{\begin{array}{|cc}s_{PWM_{out}}^{cu} & o \\ \hline -1 & 0\end{array}}{c_{PWM}}, & \text{otherwise;} \end{cases} \tag{40}$$

where

$$\psi \in (t - \Delta t_s - \Delta t, \ t - \Delta t], \tag{41}$$

is a time moment from $\Theta$ (values of $u_{PWM_{in}}$ are checked for such time moments with sampling step $\Delta t_s$, and

$$u_{Triang} : \Theta \to [0, 1] \tag{42}$$

is a triangular function with period 1 and amplitude $\Delta t_s = \frac{1}{F_s}$). The two cases of index matrix (40) determine digital signal $s_{PWM_{out}}(t)$, $t \in \Theta$:

$$u_{Triang}(t) < u_{PWM_{in}}(\psi) \Rightarrow s_{PWM_{out}}(t) = 1; \tag{43}$$

$$u_{Triang}(t) \geq u_{PWM_{in}}(\psi) \Rightarrow s_{PWM_{out}}(t) = 0. \tag{44}$$

### 3.2. Models of Component Connections and Electronic Circuits

In [1], nodes and loops are modeled according to Kirchhoff's laws. The respective sums of R-IMs in case of the present paper are

$$\overline{F_N} = \begin{array}{c|cccccc} & i_E^{cu} & i_{T_{out}}^{cu} & i_D^{cu} & i_L^{cu} & i_C^{cu} & i_R^{cu} \\ \hline n_1 & -1 & -1 & 0 & 0 & 0 & 0 \\ n_2 & 0 & 1 & 1 & -1 & 0 & 0 \\ n_3 & 0 & 0 & 0 & 1 & -1 & -1 \end{array} ; \tag{45}$$

$$\overline{F_P} = \begin{array}{c|cccccc} & u_E^{cu} & u_{T_{out}}^{cu} & u_D^{cu} & u_L^{cu} & u_C^{cu} & u_R^{cu} \\ \hline p_1 & 1 & -1 & 1 & 0 & 0 & 0 \\ p_2 & 0 & 0 & 1 & 1 & 1 & 0 \\ p_3 & 0 & 0 & 0 & 0 & -1 & 1 \end{array} ; \tag{46}$$

The first one models nodes according to Kirchhoff's current law, and the second one models loops according to Kirchhoff's voltage law. The upper R-IMs are constructed in order conditions

$$\text{rank}\left(CM_{U_N,V_N}\overline{F_N}\right) = \left|I_1\left(\overline{F_N}\right)\right| ; \qquad \text{rank}\left(CM_{U_P,V_P}\overline{F_P}\right) = \left|I_1\left(\overline{F_P}\right)\right| ; \tag{47}$$

to be satisfied, where $U_N$, $V_N$, $U_P$ and $V_P$ are permutations of the index sets of $\overline{F_N}$ or $\overline{F_P}$ (see (5) and (6)). Here and below, rank(A) denotes the rank of classical matrix A, and $|S|$ denotes the cardinality of set S.

$$\overline{F_{PCirc}}(t) \otimes \left( \overline{X_{PCirc}} \oplus \begin{array}{c|c} & o \\ \hline o & 1 \end{array} \right) = \overline{O} ,$$

$$\overline{F_{PCirc}}(t) = \overline{F_{PCs}}(t) \oplus \overline{F_N} \oplus \overline{F_P} ,$$

$$\overline{X_{PCirc}}(t) = \overline{X_{PCs}}(t) \tag{48}$$

represent the considered power circuit with its components and junctions. In order to create a more compact model, the following function of R-IM $\overline{F}(t)$ is defined below:

$$X\left(\overline{F}(t)\right) = \left[ I_2\left(\overline{F}(t)\right), \quad \{o\}, \quad \{x_{m,o}(t)\}_{m \in I_2\left(\overline{F}(t)\right)} \right]$$

$$x_{m,o}(t) = \begin{cases} u_e(t), & \text{if } m \equiv u_e^{cu}; \\ i_e(t), & \text{if } m \equiv i_e^{cu}; \\ u_e(t-\Delta t), & \text{if } m \equiv u_e^{pr}; \\ i_e(t-\Delta t), & \text{if } m \equiv i_e^{pr}; \\ s_e(t), & \text{if } m \equiv s_e^{cu}; \\ 1, & \text{if } m \equiv o. \end{cases} \tag{49}$$

For example (see (19) and (20)),

$$X\left(\overline{F_L}(t)\right) = \left( \begin{array}{c|ccc} & u_L^{cu} & i_L^{cu} & i_L^{pr} \\ \hline o & u_L(t) & i_L(t) & i_L(t-\Delta t) \end{array} \right)^T . \tag{50}$$

Indeed,

$$\left| I_2\left(\overline{F_{PCirc}}(t)\right)\right| = 14; \qquad \left| I_1\left(\overline{X_{PCirc}}(t) \oplus \begin{array}{c|c} & o \\ \hline o & 1 \end{array}\right)\right| = 25; \tag{51}$$

and 11 columns of $\overline{X_{PCirc}}(t)$ are not usable in (48). The following statements hold:

$$\overline{F_{PC}}(t) \otimes \left(\overline{X_{PC}}(t) \oplus \begin{array}{c|c} & o \\ \hline o & 1 \end{array}\right) = \overline{F_{PC}}(t) \odot X\left(\overline{F_{PC}}(t)\right) = \overline{O},$$

$$PC \in \{E, \ T_{out}, \ D, \ L, \ C, \ R\}; \tag{52}$$

$$\overline{F_{PCs}}(t) \otimes \left(\overline{X_{PCs}}(t) \oplus \begin{array}{c|c} & o \\ \hline o & 1 \end{array}\right) = \overline{F_{PCs}}(t) \odot X\left(\overline{F_{PCs}}(t)\right) = \overline{O}; \tag{53}$$

$$\overline{F_{PCirc}}(t) \otimes \left(\overline{X_{PCirc}}(t) \oplus \begin{array}{c|c} & o \\ \hline o & 1 \end{array}\right) = \overline{F_{PCirc}}(t) \odot X\left(\overline{F_{PCirc}}(t)\right) = \overline{O}. \tag{54}$$

The upper index matrices, whose identifiers start with "F", together with the considered function uniquely define the presented power circuit.

Other common statements on analog values are

$$\overline{F_{CCs}^{Analog}}(t) \odot X\left(\overline{F_{CCs}^{Analog}}(t)\right) = \overline{O}; \qquad \overline{F_{CCs}^{Analog}}(t) = \overline{F_{Sub}} \oplus \overline{F_{PI}} \tag{55}$$

The abbreviation "CCs" means "control components". Their connections (including ones with power components) which concern analog values are presented in

$$\overline{F_Q^{Analog}} = \begin{array}{c|cccccc} & u_R^{cu} & u_{Sub_{in}}^{cu} & u_{Sub_{out}}^{cu} & u_{PI_{in}}^{cu} & u_{PI_{out}}^{cu} & u_{PWM_{in}}^{cu} \\ \hline q_1^{Analog} & -1 & 1 & 0 & 0 & 0 & 0 \\ q_2^{Analog} & 0 & 0 & -1 & 1 & 0 & 0 \\ q_3^{Analog} & 0 & 0 & 0 & 0 & -1 & 1 \end{array}. \tag{56}$$

Statements

$$\overline{F_{CCirc}^{Analog}}(t) \odot X\left(\overline{F_{CCirc}^{Analog}}(t)\right) = \overline{O}; \qquad \overline{F_{CCirc}^{Analog}}(t) = \overline{F_{CCs}^{Analog}}(t) \oplus \overline{F_Q^{Analog}}; \tag{57}$$

represent the considered analog control circuit with its components and junctions in a proper manner.

R-IMs and their equations can be aggregated in

$$\overline{F_{All}^{Analog}}(t) \odot X\left(\overline{F_{All}^{Analog}}(t)\right) = \overline{O}; \qquad \overline{F_{All}^{Analog}}(t) = \overline{F_{PCirc}}(t) \oplus \overline{F_{CCirc}^{Analog}}(t). \tag{58}$$

Dependencies on digital signals are presented in

$$\overline{F_{CCs}^{Digital}}(t) = \overline{F_{PWM}}(t); \qquad \overline{F_Q^{Digital}} = \begin{array}{c|cc} & s_{PWM_{out}}^{cu} & s_{T_{in}}^{cu} \\ \hline q_1^{Digital} & -1 & 1 \end{array}; \tag{59}$$

$$\overline{F_{All}^{Digital}}(t) = \overline{F_{CCirc}^{Digital}}(t) = \overline{F_{CCs}^{Digital}}(t) \oplus \overline{F_Q^{Digital}} =$$

$$
=\begin{cases}
\overline{F_{All,1}^{Digital}} = \cfrac{c_{PWM}}{q_1^{Digital}}\left|\begin{array}{ccc} s_{PWM_{out}}^{cu} & o & s_{T_{in}}^{cu} \\ -1 & 1 & 0 \\ -1 & 0 & 1 \end{array}\right|, & \text{if } u_{Triang}(t) < u_{PWM_{in}}(\psi); \\[6pt]
\overline{F_{All,0}^{Digital}} = \cfrac{c_{PWM}}{q_1^{Digital}}\left|\begin{array}{ccc} s_{PWM_{out}}^{cu} & o & s_{T_{in}}^{cu} \\ -1 & 0 & 0 \\ -1 & 0 & 1 \end{array}\right|, & \text{otherwise;}
\end{cases}
\tag{60}
$$

under conditions (41) and (42). An R-IM equation based on $\overline{F_{All}^{Digital}}(t)$, similar to this one in (58), can be constructed:

$$
\overline{F_{All}^{Digital}}(t) \odot X\left(\overline{F_{All}^{Digital}}(t)\right) = \overline{O}.
\tag{61}
$$

R-IMs and their equations can be aggregated in

$$
\overline{F_{All}}(t) \odot X\left(\overline{F_{All}}(t)\right) = \overline{O}; \qquad \overline{F_{All}}(t) = \overline{F_{All}^{Analog}}(t) \oplus \overline{F_{All}^{Digital}}(t).
\tag{62}
$$

The upper statements form a clear mathematical model, which is solved in the next section. However, $\overline{F_{All}}(t)$ is not applied explicitly, whereas variants of $\overline{F_{All}^{Analog}}(t)$ and $\overline{F_{All}^{Digital}}(t)$ are used according to different cases.

## 4. Model Solving

### 4.1. Common Notes

$\overline{F_{All}^{Analog}}(t)$ can be divided by columns to three R-IMs in the following way:

$$
\overline{F_{All}^{Analog}}(t) = \overline{F_{cu}^{Analog}}(t) \oplus \overline{F_{pr}^{Analog}}(t) \oplus \overline{F_{o}^{Analog}}(t),
$$

$$
\overline{F_{h}^{Analog}}(t) = \text{pro}_{I_1\left(\overline{F_{All}^{Analog}}(t)\right),I_h} \overline{F_{All}^{Analog}}(t), \qquad h\in\{cu, \ pr, \ o\};
\tag{63}
$$

where $I_{cu}$, $I_{pr}$, $I_o \equiv \{o\} \subset I_2\left(\overline{F_{All}^{Analog}}(t)\right)$ are index sets, which concern all current values, all previous values and all constants, respectively. In case of the considered buck converter with a PI controller, $\overline{F_{All}^{Analog}}(t)$ has 17 rows and 23 columns:

$$
\left|I_1\left(\overline{F_{All}^{Analog}}(t)\right)\right| = 17, \qquad \left|I_2\left(\overline{F_{All}^{Analog}}(t)\right)\right| = 23.
\tag{64}
$$

$$
\left|I_2\left(\overline{F_{cu}^{Analog}}(t)\right)\right| = 17, \qquad \left|I_2\left(\overline{F_{pr}^{Analog}}(t)\right)\right| = 5, \qquad \left|I_2\left(\overline{F_{o}^{Analog}}(t)\right)\right| = |\{o\}| = 1;
\tag{65}
$$

$$
I_2\left(\overline{F_{pr}^{Analog}}(t)\right) = \left\{i_L^{pr}, \ u_C^{pr}, \ i_C^{pr}, \ u_{PI_{in}}^{pr}, \ u_{PI_{out}}^{pr}\right\};
\tag{66}
$$

i.e., all 23 columns of $\overline{F_{All}^{Analog}}(t)$ (which is sparse—more of its elements are zeros) are divided into three R-IMs with 17, 5 and 1 columns, respectively. (58) and (63) lead to

$$
\left[\overline{F_{cu}^{Analog}}(t) \odot X\left(\overline{F_{cu}^{Analog}}(t)\right)\right] \oplus \left[\overline{F_{pr}^{Analog}}(t) \odot X\left(\overline{F_{pr}^{Analog}}(t)\right)\right] \oplus \overline{F_{o}^{Analog}}(t) = \overline{O}.
\tag{67}
$$

In the last equation, the following equivalencies are used:

$$X\left(\overline{F_o^{Analog}}(t)\right) = \frac{o}{o\mid 1}; \qquad \overline{F_o^{Analog}}(t) \odot X\left(\overline{F_o^{Analog}}(t)\right) = \overline{F_o^{Analog}}(t). \tag{68}$$

Six classical matrices based on $h \in \{cu, \quad pr, \quad o\}$ can be defined (see (5) and (6)):

$$F_h^{Analog}(t) = CM_{U,V_h}\overline{F_h^{Analog}}(t), \qquad X_h^{Analog}(t) = CM_{V_h,\langle o\rangle}X\left(\overline{F_h^{Analog}}(t)\right), \tag{69}$$

where $U = \langle u_1, \quad u_2, \quad \ldots, \quad u_{|U|}\rangle$ and $V_h = \langle V_{h,1}, \quad V_{h,2}, \quad \ldots, \quad V_{h,|V_h|}\rangle$ are permutations of $I_1\left(\overline{F_{All}^{Analog}}(t)\right)$ and $I_2\left(\overline{F_{All}^{Analog}}(t)\right)$, respectively. Since $X\left(\overline{F_o^{Analog}}(t)\right)$ is a unit matrix, (67) and (69) derive the following equation:

$$F_{cu}^{Analog}(t)X_{cu}^{Analog}(t) + F_{pr}^{Analog}(t)X_{pr}^{Analog}(t) + F_o^{Analog}(t) = O. \tag{70}$$

In case of nonsingular square matrix $F_{cu}^{Analog}(t)$:

$$X_{cu}^{Analog}(t) = B'(t)X_{pr}^{Analog}(t) + J'(t),$$

$$B'(t) = -\left[F_{cu}^{Analog}(t)\right]^{-1}F_{pr}^{Analog}(t), \qquad J'(t) = -\left[F_{cu}^{Analog}(t)\right]^{-1}F_o^{Analog}(t). \tag{71}$$

A respective R-IM equation is

$$X\left(\overline{F_{cu}^{Analog}}(t)\right) = \left[\overline{B}(t) \odot X\left(\overline{F_{pr}^{Analog}}(t)\right)\right] \oplus \bar{J}(t),$$

$$\overline{B}(t) = \left[I_2\left(\overline{F_{cu}^{Analog}}(t)\right), \quad I_2\left(\overline{F_{pr}^{Analog}}(t)\right), \quad \left\{b_{u_m,v_{pr,n}} \equiv b'_{m,n}\right\}_{n\in 1,2,\ldots,|V_{pr}|}^{m\in 1,2,\ldots,|U|}\right],$$

$$\bar{J}(t) = \left[I_2\left(\overline{F_{cu}^{Analog}}(t)\right), \quad \{o\}, \quad \left\{b_{u_m,o} \equiv j'_{m,1}\right\}_{m\in 1,2,\ldots,|U|}\right]. \tag{72}$$

According to (65) in the discussed example, $\overline{B}(t)$ has 17 rows and 5 columns, and $\bar{J}(t)$ has 17 rows and 1 column. They uniquely define all voltages and amperages of analog components. Index sub-matrices can be used, too:

$$pro_{I,\{o\}}X\left(\overline{F_{cu}^{Analog}}(t)\right) = \left[\overline{B_I}(t) \odot X\left(\overline{F_{pr}^{Analog}}(t)\right)\right] \oplus \overline{J_I}(t), \qquad I \subseteq I_2\left(\overline{F_{cu}^{Analog}}(t)\right)$$

$$\overline{B_I}(t) = pro_{I,I_2(\overline{B}(t))}\overline{B}(t), \qquad \overline{J_I}(t) = pro_{I,\{o\}}\bar{J}(t), \tag{73}$$

Such an approach is basic in one of the presented algorithms, in which I depend on

$$I_{cu,1}^{Analog} \equiv \left\{u_e^{cu} \mid u_e^{pr} \in I_2\left(\overline{F_{pr}^{Analog}}(t)\right)\right\} \cup \left\{i_e^{cu} \mid i_e^{pr} \in I_2\left(\overline{F_{pr}^{Analog}}(t)\right)\right\}; \tag{74}$$

this index set denotes all current values, whereas the respective previous ones are included in (69)–(73).

$\overline{B_I}(t)$, $\overline{J_I}(t)$, $\overline{F_{All}^{Analog}}(t)$ and many others from the upper R-IMs depend on time. Alternative constant index matrices can be constructed; such ones are described in (29),

(32), (40) and (60). Index set $I_{cu,2}^{Analog}$ can be defined as related to values in conditions on time-dependent R-IMs in case of the considered converter

$$I_{cu,2}^{Analog} = \left\{ u_D^{cu}, \quad u_{T_{out}}^{cu}, \quad u_{PWM_{in}}^{cu} \right\} \tag{75}$$

A scheme according to the conditions is given below:

$$\left( s_{T_{in}}(t) = 0 \lor u_{T_{out}}(t) \leq U_{T_{out}}^0 \right) \land u_D(t) \leq U_D^0 \Leftrightarrow \overline{F_{T_{out}}}(t) = \overline{F_{T_{out}}^0} \land$$

$$\land \overline{F_D}(t) = \overline{F_D^0} \Leftrightarrow \overline{F_{All}^{Analog}}(t) \equiv \overline{F_{All,0,0}^{Analog}} \supseteq \overline{F_{T_{out}}^0} \land \overline{F_{All,0,0}^{Analog}} \supseteq \overline{F_D^0}; \tag{76}$$

$$\left( s_{T_{in}}(t) = 0 \lor u_{T_{out}}(t) \leq U_{T_{out}}^0 \right) \land u_D(t) > U_D^0 \Leftrightarrow \overline{F_{T_{out}}}(t) = \overline{F_{T_{out}}^0} \land$$

$$\land \overline{F_D}(t) = \overline{F_D^1} \Leftrightarrow \overline{F_{All}^{Analog}}(t) \equiv \overline{F_{All,0,1}^{Analog}} \supseteq \overline{F_{T_{out}}^0} \land \overline{F_{All,0,1}^{Analog}} \supseteq \overline{F_D^1}; \tag{77}$$

$$\left( s_{T_{in}}(t) = 1 \land u_{T_{out}}(t) > U_{T_{out}}^0 \right) \land u_D(t) \leq U_D^0 \Leftrightarrow \overline{F_{T_{out}}}(t) = \overline{F_{T_{out}}^1} \land$$

$$\land \overline{F_D}(t) = \overline{F_D^0} \Leftrightarrow \overline{F_{All}^{Analog}}(t) \equiv \overline{F_{All,1,0}^{Analog}} \supseteq \overline{F_{T_{out}}^1} \land \overline{F_{All,1,0}^{Analog}} \supseteq \overline{F_D^0}; \tag{78}$$

$$\left( s_{T_{in}}(t) = 1 \land u_{T_{out}}(t) > U_{T_{out}}^0 \right) \land u_D(t) > U_D^0 \Leftrightarrow \overline{F_{T_{out}}}(t) = \overline{F_{T_{out}}^1} \land$$

$$\land \overline{F_D}(t) = \overline{F_D^1} \Leftrightarrow \overline{F_{All}^{Analog}}(t) \equiv \overline{F_{All,1,1}^{Analog}} \supseteq \overline{F_{T_{out}}^1} \land \overline{F_{All,1,1}^{Analog}} \supseteq \overline{F_D^1} \tag{79}$$

(relation "$\supseteq$" on index matrices is presented in (9)). The upper variants of $\overline{F_{All}^{Analog}}(t)$ (which are $\overline{F_{All,0,0}^{Analog}}$, $\overline{F_{All,0,1}^{Analog}}$, $\overline{F_{All,1,0}^{Analog}}$ and $\overline{F_{All,1,1}^{Analog}}$ ) are constructed in order for all the functioning modes to be handled via respective variants of $\overline{B_I}(t)$ and $\overline{J_I}(t)$ through (73) (all these calculations can be performed in advance).

$$\overline{F_{All}^{Digital}}(t) \odot X\left( \overline{F_{All}^{Digital}}(t) \right) = \overline{O} \tag{80}$$

describes digital signals. The common statement

$$X\left( \overline{F_{cu}^{Digital}}(t) \right) = IM_{V_{cu},\{o\}} \left[ -\left( CM_{U,V_{cu}} \overline{F_{cu}^{Digital}}(t) \right)^{-1} CM_{U,\{o\}} \overline{F_o^{Digital}}(t) \right]; \tag{81}$$

Determines all digital values; here, U and $V_{cu}$ are permutations of the first and the second index sets of $\overline{F_{cu}^{Digital}}(t)$, respectively ($\overline{F_{cu}^{Digital}}(t)$ and $\overline{F_o^{Digital}}(t)$ are defined similarly to these ones in (63)).

### 4.2. Model-Solving Algorithm

An algorithm (Algorithm 1), which is presented in case of the considered example, is given below. Its first three steps are implemented once, and its next three steps are performed cyclically until statement $t \leq t_{end}$ holds.

---

**Algorithm 1: Single simulation of device functioning**

---

Step 1: Specifying a time interval and a time step

Let the functioning of the model be simulated in time set (18) with a proper initial time $t_0$ (for a convenience, it is set to zero below) and a proper time step $\Delta t$. It is appropriate for the end model time $t_{end}$ to be used, too. The choice of $\Delta t$ determines a proper accuracy; Figure 2 shows the results of simulations depending on this choice (relative to a switching interval).

Step 2: Initialization

Initial values corresponding to index sets $I_{cu,1}^{Analog}$ and $I_{cu,2}^{Analog}$ must be set. In case of zero values of voltages and amperages at $t_0$ for the power components of the considered DC-DC converter, the following R-IM can be used:

$$\text{pro}_{I_{cu,1}^{Analog} \cup I_{cu,2}^{Analog}, \{o\}} X\left(\overline{F_{cu}^{Analog}}(t_0)\right) =$$

$$= \left(\begin{array}{c|cccccccc} & u_D^{cu} & u_{T_{out}}^{cu} & i_L^{cu} & u_C^{cu} & i_C^{cu} & u_{PI_{in}}^{cu} & u_{PI_{out}}^{cu} & u_{PWM_{in}}^{cu} \\ \hline o & 0 & 0 & 0 & 0 & 0 & U_{Target} & K_P U_{Target} & K_P U_{Target} \end{array}\right)^T; \tag{82}$$

Here, $u_{PI_{out}}(t_0) = u_{PWM_{in}}(t_0)$ holds.

Step 3: Specifying constant R-IMs

Such R-IMs are calculated in advance and explored in the next steps.

Four Variants of $\overline{F_{All}^{Analog}}(t)$ are described in (76)–(79). The following variants of index matrices from (73) can be presented:

$$\overline{B_{I,m,n}}, \qquad \overline{J_{I,m,n}}, \qquad m \in \{0,1\}, \qquad n \in \{0,1\},$$

$$I \in \left\{I_{cu,2}^{Analog}, \quad I'\right\}, \qquad I_{cu,1}^{Analog} - I_{cu,2}^{Analog} \subseteq I' \subseteq I_2\left(\overline{F_{cu}^{Analog}}(t)\right) - I_{cu,2}^{Analog}; \tag{83}$$

$\overline{B_{I,m,n}}$ and $\overline{J_{I,m,n}}$ are variants of $\overline{B_I}$ and $\overline{J_I}$ respectively in case of

$$\overline{F_{All}^{Analog}}(t) = \overline{F_{All,m,n}^{Analog}} \tag{84}$$

($I'$ contains all indexes related to analog values which should be investigated except for those values which are related to $I_{cu,2}^{Analog}$; the last ones are calculated separately).

Two variants of $\overline{F_{All}^{Digital}}(t)$ are described in (60). The following variants of index matrix $X\left(\overline{F_{cu}^{Digital}}(t)\right)$ can be calculated in advance according to (60) and (81):

$$\overline{X_{cu,1}^{Digital}} = IM_{\langle s_{PWM_{out}}^{cu}, \quad s_{T_{in}}^{cu}\rangle, \langle o\rangle}\left[-\begin{pmatrix} -1 & 0 \\ -1 & 1 \end{pmatrix}^{-1}\begin{pmatrix} 1 \\ 0 \end{pmatrix}\right] = \begin{array}{cc|c} & & o \\ \hline \overset{cu}{s_{PWM_{out}}} & & 1 \\ s_{T_{in}}^{cu} & & 1 \end{array}; \tag{85}$$

$$\overline{X_{cu,0}^{Digital}} = IM_{\langle s_{PWM_{out}}^{cu}, \quad s_{T_{in}}^{cu}\rangle, \langle o\rangle}\left[-\begin{pmatrix} -1 & 0 \\ -1 & 1 \end{pmatrix}^{-1}\begin{pmatrix} 0 \\ 0 \end{pmatrix}\right] = \begin{array}{cc|c} & & o \\ \hline \overset{cu}{s_{PWM_{out}}} & & 0 \\ s_{T_{in}}^{cu} & & 0 \end{array}; \tag{86}$$

(trivial equivalence $s_{PWM_{out}}(t) = s_{T_{in}}(t)$ holds).

Step 4: Current time setting

Add $\Delta t$ to current time $t$.

Step 5: Calculation of digital values

Generally, these values depend on analog and other digital values. All described analog values are continuous; small values of $\Delta t$ determine near their values, which can be used like replaceable ones. An interrelation between digital values has been described in (81). For the considered converter, the signal at the input of the presented PWM block is continuous, and $u_{PWM_{in}}(\psi)$ for current time $t$ is modeled by either $u_{PWM_{in}}(t-\Delta t)$ or an earlier value (see (41), where sampling moment $\psi$ depends on triangular function (42)). The latter and $u_{PWM_{in}}(\psi)$ determine $\overline{F_{All}^{Digital}}(t)$ in (60) and, therefore, all digital values through (85) or (86).

---

---

**Algorithm 1** *Cont.*

---

Step 6: Calculation of analog values

Analog values which concern time $t-\Delta t$, as near to respective values for time t, have been used in the previous step. Oppositely, analog values at time t, which are related to conditions of other values (and, therefore, indexed in $I_{cu,2}^{Analog}$ ), are used directly in this step in order for variants of $\overline{F_{All}^{Analog}}(t)$ to be chosen. For the considered converter, $I_{cu,2}^{Analog}$ has three elements. For example, if for the analog values at time t, which are indexed in $I_{cu,2}^{Analog}$ and estimated by

$$
\begin{array}{c|c}
 & o \\
\hline
u_D^{cu} & u_D(t) \\
u_{T_{out}}^{cu} & u_{T_{out}}(t) \\
u_{PWM_{in}}^{cu} & u_{PWM_{in}}(t)
\end{array}
= \left[ \overline{B_{I_{cu,2}^{Analog},0,0}} \odot X\left( \overline{F_{pr}^{Analog}}(t) \right) \right] \oplus \overline{J_{I_{cu,2}^{Analog},0,0}}, \tag{87}
$$

the inequalities in (76) hold, then the following assignments must be implemented:

$$
\overline{B_{I'}}(t) \equiv \overline{B_{I',0,0}}, \qquad \overline{J_{I'}}(t) \equiv \overline{J_{I',0,0}}; \tag{88}
$$

See (83) and (84) for the upper notations (here, $\overline{F_{All}^{Analog}}(t) \equiv \overline{F_{All,0,0}^{Analog}}$ ). Similar statements can be presented according to (77)–(79). Since all considered analog values are continuous, only one of the cases (76)–(79) can be applied $\overline{B_{I'}}(t)$ and $\overline{J_{I'}}(t)$ are used in the following calculations. In case of $I' \equiv I_{cu,1}^{Analog} - I_{cu,2}^{Analog} \equiv I_{cu,1}^{Analog}$:

$$
\begin{array}{c|c}
 & o \\
\hline
i_L^{cu} & i_L(t) \\
u_C^{cu} & u_C(t) \\
i_C^{cu} & i_C(t) \\
u_{PI_{in}}^{cu} & u_{PI_{in}}(t) \\
u_{PI_{out}}^{cu} & u_{PI_{out}}(t)
\end{array}
= \left( \overline{B_{I_{cu,1}^{Analog}}}(t) \odot
\begin{array}{c|c}
 & o \\
\hline
i_L^{pr} & i_L(t-\Delta t) \\
u_C^{pr} & u_C(t-\Delta t) \\
i_C^{pr} & i_C(t-\Delta t) \\
u_{PI_{in}}^{pr} & u_{PI_{in}}(t-\Delta t) \\
u_{PI_{out}}^{pr} & u_{PI_{out}}(t-\Delta t)
\end{array}
\right) \oplus \overline{J_{I_{cu,1}^{Analog}}}(t). \tag{89}
$$

Since $u_C(t) \equiv u_R(t)$ the output voltage of the considered DC-DC buck converter with a PI controller can be investigated. In case of $I' \equiv I_2\left( \overline{F_{cu}^{Analog}}(t) \right) - I_{cu,2}^{Analog}$, all analog values can be investigated. On each iteration, elements of $I'$ can be modified under constraints in (83) in order for proper sets of current voltages and amperages to be estimated.

Step 7: Time check

If $t + \Delta t \leq t_{end}$ holds, then Steps 4, 5 and 6 must be performed; otherwise, the algorithm must be terminated.

---

### 4.3. Single Simulation Results

Figure 2 presents output voltage simulation results for the considered converter. Algorithm 1 is implemented through MATLAB source code (without Simulink to be used). In case of Figure 2b–d, the described model varies on a sampling frequency (relative to a switching frequency). A comparison with the MATLAB Simulink model (see Figure 2a) has been made. The simulations are performed with the following parameters:

$$
E = 20 \text{ V}; \qquad U_{Target} = 10 \text{ V}; \qquad R = 100 \ \Omega; \qquad F_s = 400 \text{ kHz}; \tag{90}
$$

$$
R_E = 100 \text{ m}\Omega; \qquad R_L = 20 \text{ m}\Omega; \qquad R_C = 10 \text{ m}\Omega; \tag{91}
$$

$$
U_{T_{out}}^0 = 500 \text{ mV}; \qquad R_T = 20 \text{ m}\Omega; \qquad U_D^0 = 500 \text{ mV}; \qquad R_D = 50 \text{ m}\Omega; \tag{92}
$$

$$
L = 10 \ \mu\text{H}; \qquad C = 5 \text{ mF}; \qquad K_p = 200; \qquad K_i = 200. \tag{93}
$$

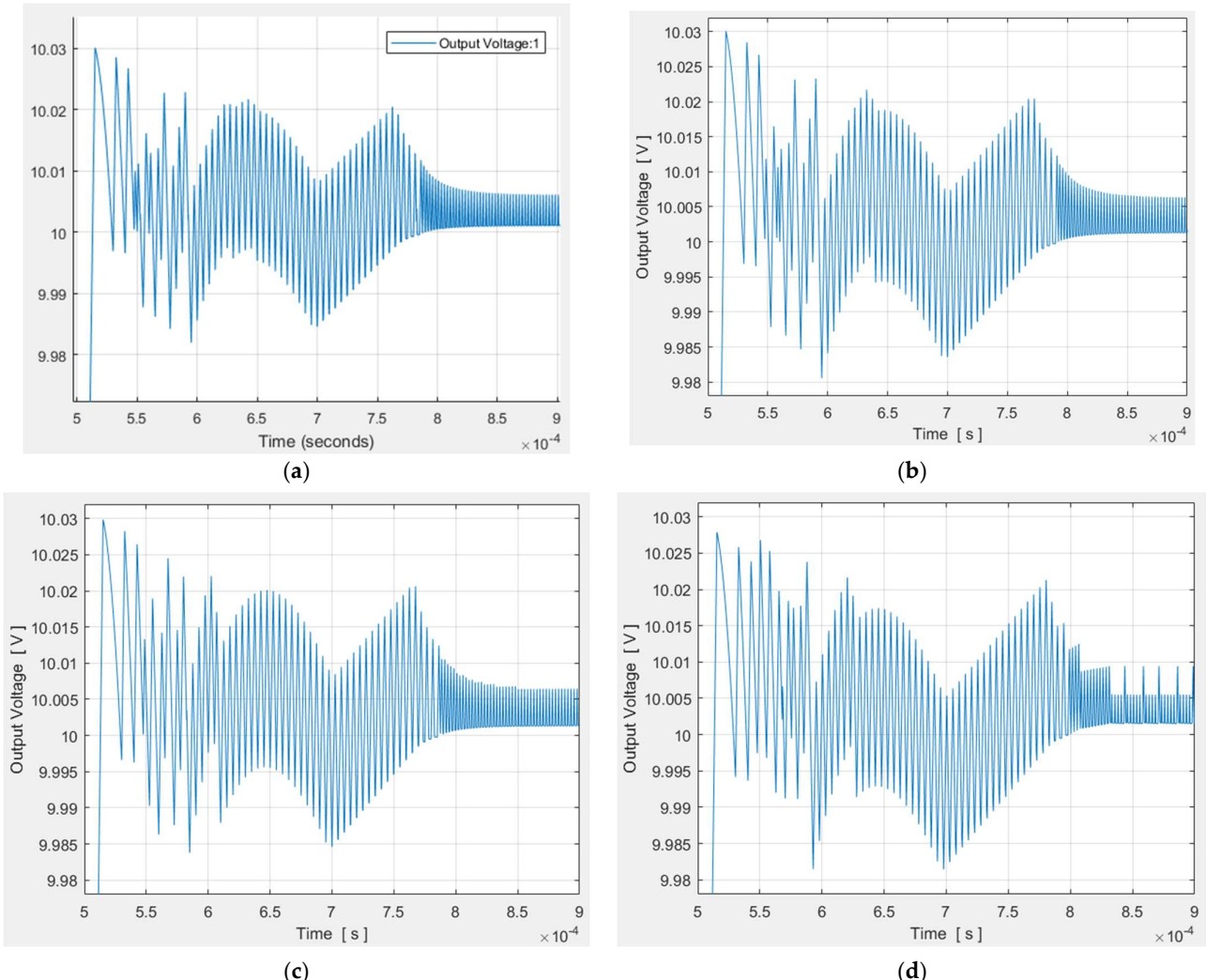

**Figure 2.** Output voltage simulation results for Simulink and the proposed R-IM-based software. Modeling with (**a**) Simulink; (**b**) R-IM-based software with ratio $F_d / F_s = 600$; (**c**) R-IM-based software with ratio $F_d / F_s = 60$; (**d**) R-IM-based software with ratio $F_d / F_s = 6$.

The full meaning of these parameters is given in (19), (26), (29), (32), (35) and (38)–(40); the uncommented ones are as follows: $U_{Target}$—the output target voltage; $K_p$ and $K_i$—the coefficients for proportional and integral terms in a PI controller.

A step response on the output is presented on Figure 2. The last one displays the fact that large changes in the $F_d / F_s$ ratio keep this step response almost unchanged. Initial values from (82) and time $t_{end} = 0.9$ ms are used. The graphs are zoomed (see whole graphs in Figure 3) in order for the similarity between cases to be shown more clearly.

Characteristics of the output signal are defined below. The maximum

$$U^s_{R,max} = \max_{\theta \in [0, \ t_{end}]} u^s_R(\theta) \tag{94}$$

(s denotes the performed simulation) for the whole time interval and the deviation

$$\Delta_{end} U^s_R = \Delta U^s_R(0.9 t_{end}) \tag{95}$$

for an end part of this interval are estimated (notation

$$\Delta U^s_R(t) = \max_{\theta \in [t, \ t_{end}]} \left| u^s_R(\theta) - U_{Target} \right| \tag{96}$$

is used here). Statements

$$U^s_{R,max} < 0.03 \text{ V}; \qquad \Delta_{end} U^s_R < 0.01 \text{ V}; \qquad (97)$$

hold in all cases, which are shown in this subsection.

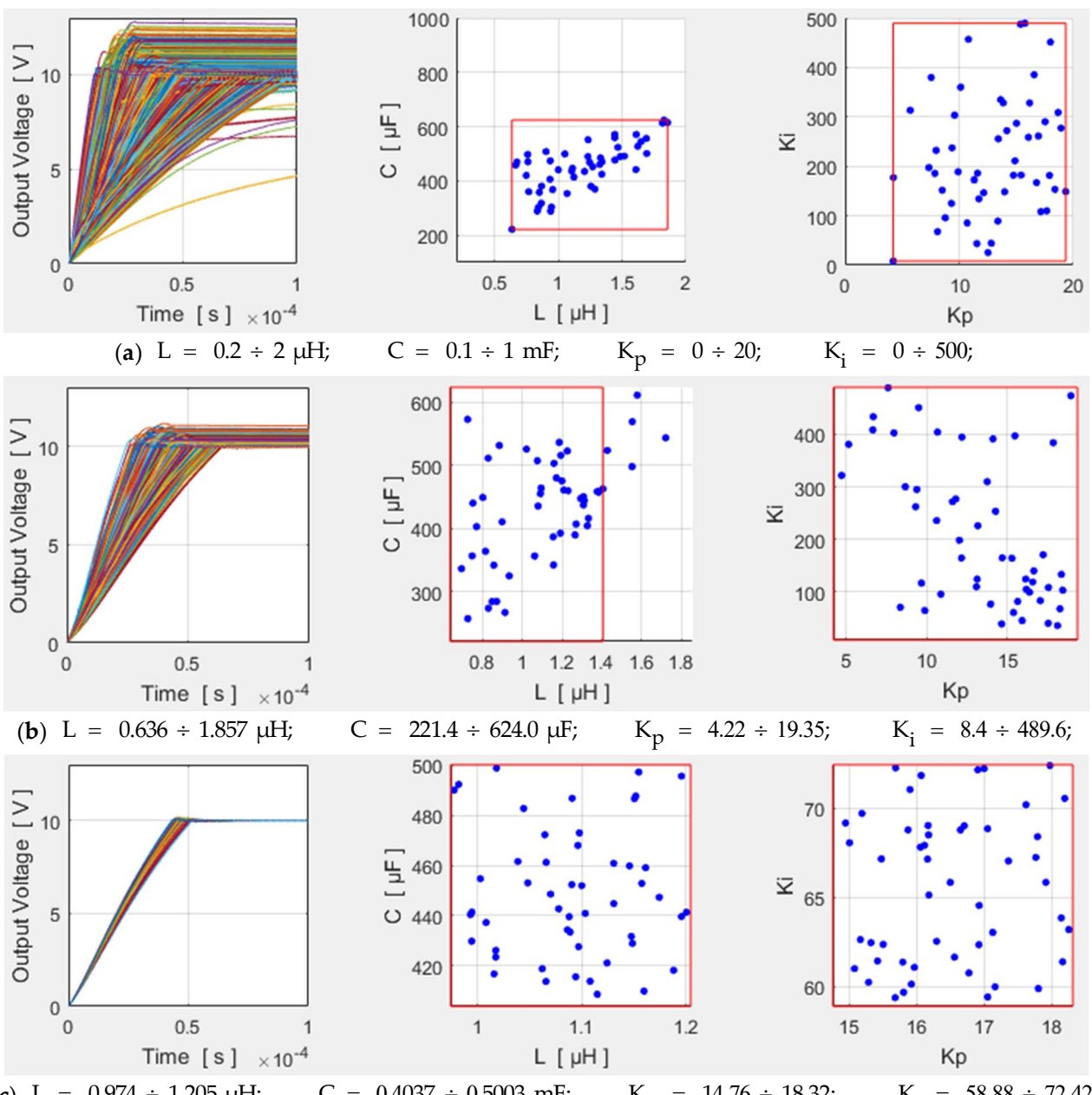

(**a**)  L = 0.2 ÷ 2 µH;    C = 0.1 ÷ 1 mF;    K_p = 0 ÷ 20;    K_i = 0 ÷ 500;

(**b**)  L = 0.636 ÷ 1.857 µH;    C = 221.4 ÷ 624.0 µF;    K_p = 4.22 ÷ 19.35;    K_i = 8.4 ÷ 489.6;

(**c**)  L = 0.974 ÷ 1.205 µH;    C = 0.4037 ÷ 0.5003 mF;    K_p = 14.76 ÷ 18.32;    K_i = 58.88 ÷ 72.42;

**Figure 3.** Simulation results for an output voltage, parameter distributions (for "successful" cases) and nested parameters. Intervals in an optimization procedure: (**a**) at the first iteration; (**b**) at the second iteration; (**c**) at the 39th iteration.

## 5. Optimization

An iterative algorithm, which determines parameter intervals in order for the proper output voltage requirements to be met, is given below (Algorithms 2 and 3).

---

**Algorithm 2: Determining final parameter intervals at once**

---

Step 1: Specifying fixed parameters

Let the input voltage, the target voltage, the output resistance, the switching frequency and the simulation duration be fixed through

$$E = 20\,\text{V}; \qquad U_{\text{Target}} = 10\,\text{V}; \qquad R = 100\,\Omega; \qquad F_s = 1\,\text{MHz}; \qquad t_{\text{end}} = 0.11\,\text{ms}; \qquad (98)$$

(in this case, $0.9t_{\text{end}} \approx 0.1\,\text{ms}$), and let the additional parameters in (91) and (92) have the same constant values.

Step 2: Specifying output voltage requirements

For the described DC-DC converter, the following requirements are taken (see (94)–(96)):

$$U^s_{R,\text{max}} < 0.2\,\text{V}; \qquad \Delta_{\text{end}} U^s_R < 0.01\,\text{V}; \qquad (99)$$

Step 3: Specifying initial parameter intervals

Let the parameters from (93) vary in order for minimal values of $U^s_{R,\text{max}}$ and $\Delta_{\text{end}} U^s_R$ to be obtained, and let their initial values be in the following ranges:

$$L = 0.2 \div 2\,\mu\text{H}; \qquad C = 0.1 \div 1\,\text{mF}; \qquad K_p = 0 \div 20; \qquad K_i = 0 \div 500. \qquad (100)$$

Step 4: Simulations with constant parameter intervals

Simulations, which use random uniformly distributed values according to (100), must be implemented. They are performed until the number of "successful" ones (in this context, the requirements from (99)) is lower than $n_{\text{Successful}}$. In the present paper,

$$n_{\text{Successful}} = 50 \qquad (101)$$

(see Figure 3). In case of the initial intervals, the number of all simulations is 939 (the results are presented in Figure 3a). Nested parameter intervals have been used in this algorithm, and the number of simulations presented in Figure 3b is 214 (the number of "successful" ones is equal to $n_{\text{Successful}}$ too). The next respective numbers of simulations are 185, 178 and 113. The 23rd simulation has 50; i.e., it is equal to $n_{\text{Successful}}$ However, 38 variants of parameter intervals are used in order for 10% parameter tolerances to be obtained (see (108)–(110)). Distributions for "successful" simulations are presented in the second and third columns in Figure 3.

Step 5: Determining nested intervals

The following intervals can be defined:

$$[\omega_{\text{min}}, \quad \omega_{\text{max}}], \qquad \omega \in \Omega \equiv \{C, \quad L, \quad K_i, \quad K_p\} \qquad (102)$$

The intervals are updated according to "successful" simulations only. Let

$$\langle \omega_1, \omega_2, \ldots, \omega_{n_{\text{Successful}}} \rangle, \qquad (i < j \Rightarrow \omega_i \leq \omega_j, \qquad i,j \in 1,2,\ldots,n_{\text{Successful}}) \qquad (103)$$

present the values of parameter $\omega$ in such simulations in an ordered manner.

In case of the initial ranges, which are presented in (100) and Figure 3a, new intervals are formed by

$$[\omega_{\text{min}}, \quad \omega_{\text{max}}] = [\omega_1, \quad \omega_{n_{\text{Successful}}}], \qquad \omega \in \Omega \equiv \{C, \quad L, \quad K_i, \quad K_p\}. \qquad (104)$$

They are shown with red lines in Figure 3a and are used in the next iteration (see the ranges of the parameters in Figure 3b).

---

---

**Algorithm 2** *Cont.*

---

In case of the next (non-initial) intervals, a single element $\mu$ from $\Omega$ is selected in order for the new interval $\begin{bmatrix} \mu_{min}, & \mu_{max} \end{bmatrix}$ to be set (the ranges of the other parameters remain the same). The percentage of values $\mu_1, \mu_2, \ldots, \mu_{n_{Successful}}$, which do not belong to $\begin{bmatrix} \mu_{min}, & \mu_{max} \end{bmatrix}$, is denoted by $\lambda$. In the present consideration, $\lambda$ is constant:

$$\lambda = 10\%. \tag{105}$$

One of the endpoints of the respective parameter interval $\begin{bmatrix} \mu_{min}^{old}, & \mu_{max}^{old} \end{bmatrix}$ from the previous iteration is updated in $\begin{bmatrix} \mu_{min}, & \mu_{max} \end{bmatrix}$. A pair containing an interval and a ratio is defined for each $\omega \in \Omega$:

$$\langle [\omega_{min}, \ \omega_{max}], \zeta\omega \rangle = \begin{cases} \left\langle \left[ \omega_{min}^{old}, \ \omega_{|(1-\lambda)n_{Successful}|} \right], \zeta_{\omega}^{begin} \right\rangle, & \text{if } \zeta_{\omega}^{begin} < \zeta_{\omega}^{end}; \\ \left\langle \left[ \omega_{|\lambda n_{Successful}+1|}, \ \omega_{max}^{old} \right], \zeta_{\omega}^{end} \right\rangle, & \text{otherwise}; \end{cases}$$

$$\zeta_{\omega}^{begin} = \frac{\omega_{|(1-\lambda)n_{Successful}|} - \omega_{min}^{old}}{\omega_{max}^{old} - \omega_{min}^{old}}, \ \zeta_{\omega}^{end} = \frac{\omega_{max}^{old} - \omega_{|\lambda n_{Successful}+1|}}{\omega_{max}^{old} - \omega_{min}^{old}}; \tag{106}$$

where $|x|$ denotes the integer part of x. Parameter $\mu$ is selected in order for the following equality to be satisfied:

$$\zeta\mu = \min_{\omega \in \Omega \wedge \psi_\omega > \frac{1+\sigma}{1-\sigma}} \zeta\omega. \tag{107}$$

Inequality $\psi_\omega > \frac{1+\sigma}{1-\sigma}$ is discussed in Step 6. $L_{max}$ is updated at the second iteration to 1.405 µH (since $\lambda = 10\%$ of 50 is equal to 5, the parameter range is reduced on the base of the highest 5 values on induction—see the second graph on Figure 3b).
Step 6: Termination check

In practice, the parameters can be realized with given tolerances. In the present paper,

$$\sigma = 10\% \tag{108}$$

is the used tolerance for each parameter; it means that only ranges, which satisfy

$$\psi_\omega = \frac{\omega_{max}}{\omega_{min}} > \frac{1+\sigma}{1-\sigma} = \frac{1.1}{0.9} \approx 1.22, \ \omega \in \Omega \equiv \{C, L, K_i, K_p\}; \tag{109}$$

Are applied. Otherwise, if inequality (109) does not hold for any new interval in Step 4, the algorithm is terminated. For the presented example, after the last allowed parameter changes, the ranges of the parameters (see Figure 3c) are

$$L = 0.974 \div 1.205 \ \mu H; \quad C = 0.4037 \div 0.5003 \ mF; \quad K_p = 14.76 \div 18.32; \quad K_i = 58.88 \div 72.42; \tag{110}$$

---

---

**Algorithm 3: Optimizing final parameter intervals**

---

Step 1: Setting a number of simulation series and a current simulation series index

    A number of simulation series m in the present paper has been set by

$$m = 10 \tag{111}$$

in order for sufficiently distinguishable graphics to be obtained (see Figure 4b,c). The current simulation series index is initialized by

$$i = 1 \tag{112}$$

Step 2: Applying an algorithm with a fixed input voltage and a fixed output resistance

    Algorithm 2 must be applied.

Step 3: Verifying results

    The final parameter intervals obtained in Step 2 through Algorithm 2 have been used here. Normal distributions based on the final parameter intervals (notation $[\omega_{min}, \quad \omega_{max}]$ from (102) are used here) with means and standard deviations

$$\mu_{\omega} = \frac{\omega_{max} + \omega_{min}}{2}, \qquad \sigma_{\omega} = \frac{\omega_{max} - \omega_{min}}{6}, \qquad \omega \in \Omega \equiv \{C, \quad L, \quad K_i, \quad K_p\} \tag{113}$$

respectively, must be constructed and applied. $\sigma_{\omega}$ ensures 99.7% of the data within $[\omega_{min}, \omega_{max}]$. Simulations with a fixed input voltage and a fixed output resistance are performed in order levels

$$\Delta U_R^{S_i}(\theta) = \max_{s \in S_i \ \equiv \{s_{i,1}, s_{i,2}, \ldots, s_{i,n}\}} \Delta U_R^s(\theta), \qquad \theta \in [t_0, \quad 0.9t_{end}] \tag{114}$$

under notation (96) to be obtained ($s_{i,1}, s_{i,2}, \ldots, s_{i,n}$ are separate simulations); in the present paper,

$$n = 10000 \tag{115}$$

    The magenta dotted line on Figure 4a presents the resulting curve from (114).

Step 4: Increasing the current number of simulation series

    Increment the current simulation series index by 1.

Step 5: A check on the current number of simulation series

    If $i \leq m$ holds, then Steps 2, 3 and 4 must be performed; otherwise, the execution of these steps must be terminated.

Step 6: Choosing an optimal parameter intervals

    It is obvious that m simulation series have been performed in the previous steps. Different levels (114) have been obtained. Optimal final parameter intervals, which correspond to a proper

$$k \in \{1, 2, \ldots, m\} \tag{116}$$

based on the simulation series with the lowest level

$$\Delta U_R^{S_k}(0.9t_{end}) = \min_{i \in \{1,2,\ldots,m\}, U_{R,max}^{S_i} \leq U_{R,max}^{fixed}} \Delta U_R^{S_i}(0.9t_{end}) \tag{117}$$

must be chosen; here

$$U_{R,max}^{S_i} = \max_{s \in S_i \ \equiv \{s_{i,1}, s_{i,2}, \ldots, s_{i,n}\}} U_{R,max}^s \tag{118}$$

---

| **Algorithm 3** *Cont.* |
| --- |

In the present paper

$$U_{R,max}^{fixed} = 10.2 \text{ V}. \tag{119}$$

This process is shown in Figure 4b,c, where 10 lines (one continuous and nine dotted ones) present variants of the dotted curve from Figure 4a. The red ones represent simulations, for which the following statement holds:

$$U_{R,max}^{S_i} > U_{R,max}^{fixed}. \tag{120}$$

The blue ones represent simulations, for which

$$U_{R,max}^{S_i} \leq U_{R,max}^{fixed} \tag{121}$$

and the line, which corresponds to the simulations with an optimal parameter intervals (see (117)), is continuous.

Generally, the discussed example gives the following constraints:

$$U_{R,max}^{S_i} \leq U_{R,max}^{fixed}; \qquad \Delta U_R^{S_{i'}}(0.9t_{end}) < 0.01 \text{ V} \tag{122}$$

in case of 10,000 simulations for normally distributed parameters from intervals with at least 10% tolerance for L, C, $K_P$ and $K_i$.

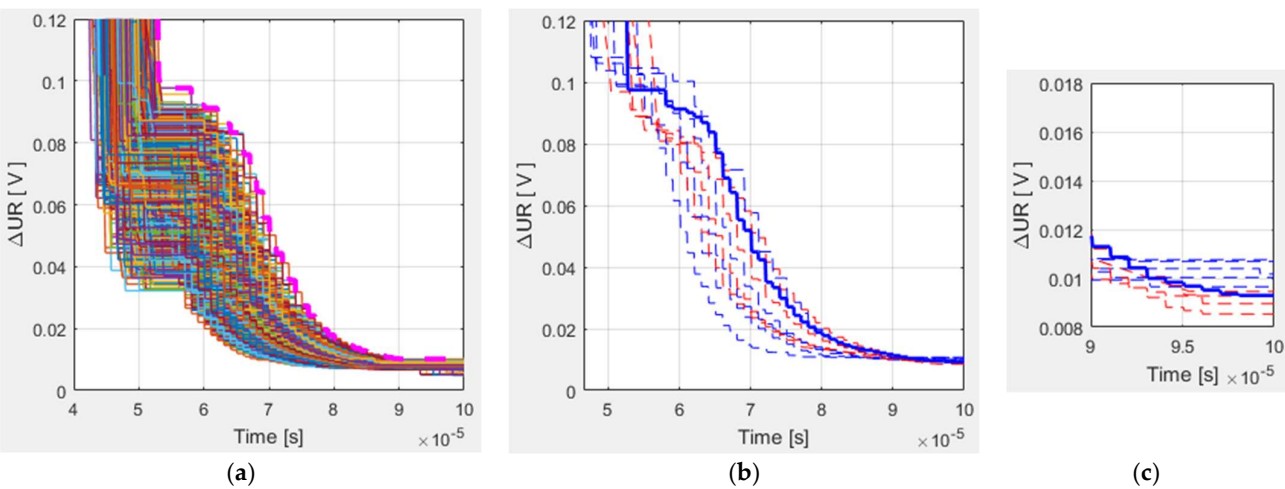

**Figure 4.** Tolerance simulation results according to Algorithm 3: (**a**) on Step 3; (**b**) on Step 6; (**c**) on Step 6 in a zoom mode.

The final parameter intervals have been obtained with a fixed input voltage of 20 V and a fixed output resistance of 100 Ω. Such intervals (based on these two values) are used below, but different input voltages and output resistances are used in Algorithm 3. Graphs of characteristic $\Delta U_R^{S_k}(0.9t_{end})$ depending on the last two parameters (with one constant parameter) are given in Figure 5.

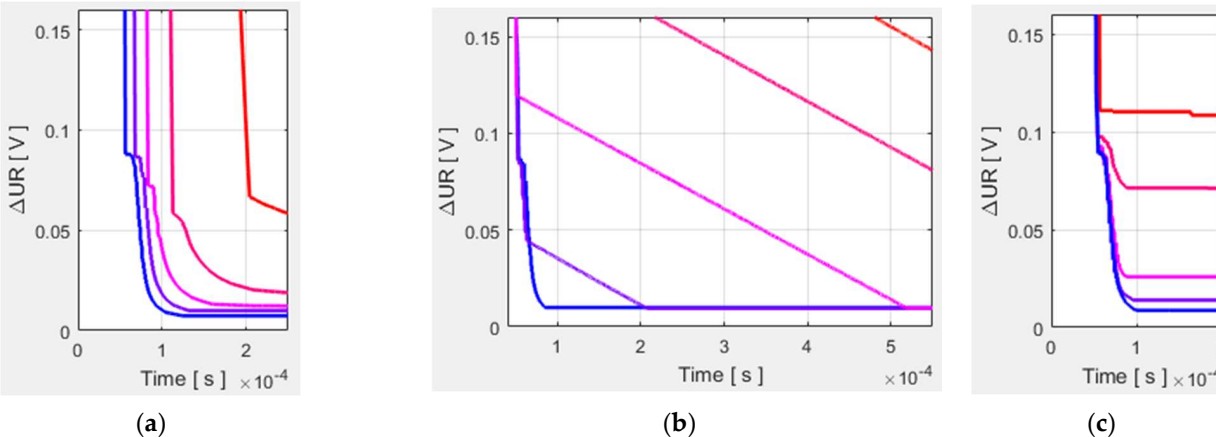

**Figure 5.** Output voltage tolerances for various input voltages or output resistances: (**a**) E = 11, 13, 15, 17, 19 V; R = 100 Ω (colored from red to blue); (**b**) E = 21, 23, 25, 27, 29 V; R = 100 Ω (colored from blue to red); (**c**) E = 20 V; R = 1, 3, 10, 30, 100 Ω (colored from red to blue).

Algorithm 3 can be modified in order for various input voltages and output resistances to be used simultaneously, too. In this case, random normally distributions on these parameters with means and standard deviations

$$\mu_\omega = \frac{\omega_{max} + \omega_{min}}{2}, \qquad \sigma_\omega = \frac{\omega_{max} - \omega_{min}}{6}, \qquad \omega \in \Omega \equiv \{E, \ R\}, \qquad (123)$$

can be used in Step 3 of the algorithm. The respective results are presented in Figure 6.

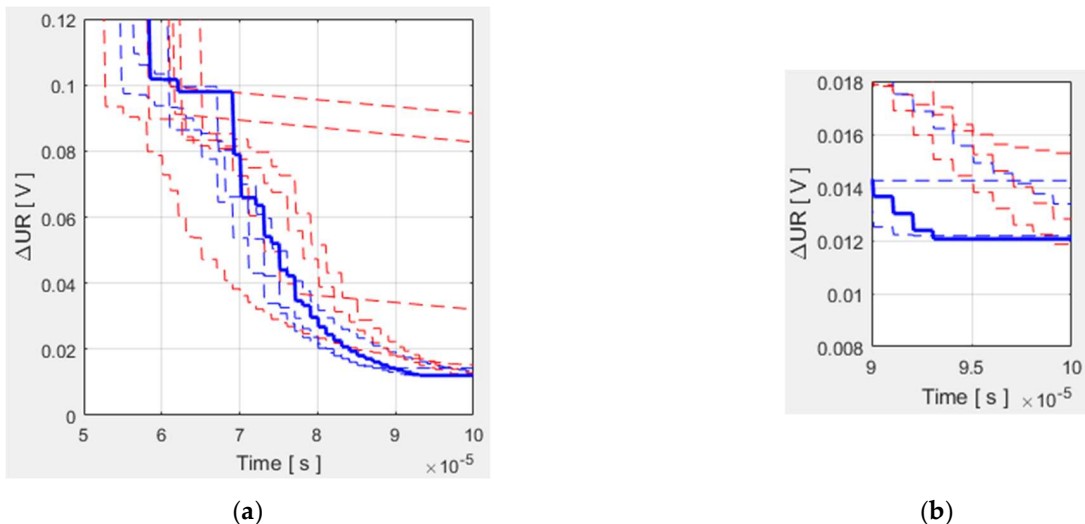

**Figure 6.** Tolerance simulation results according to Algorithm 3 with various input voltages and various output resistances (E = 18 ÷ 22 V; R = 50 ÷ 150 Ω) (**a**) on Step 6; (**b**) on Step 6 in a zoom mode.

The notations in Figure 6a are the same as the ones in Figure 4b where 10 lines (one continuous and nine dotted ones) present 10 variants. Since three graphs in Figure 6a determine tolerances on $\Delta U_R^{S_i}(0.9t_{end})$ greater than 0.018 V, only seven graphs are presented in Figure 6b. The discussed example gives the following constraints:

$$U_{R,max}^{S_k} < 10.2 \text{ V}; \qquad \Delta U_R^{S_{i'}}(0.9t_{end}) \approx 0.012 \text{ V}. \qquad (124)$$

also in case of 10,000 simulations. Normal distributions of the input voltage and the output resistance under conditions (123) with parameters defined by

$$E_{min} = 18 \text{ V}; \qquad E_{max} = 22 \text{ V}; \qquad R_{min} = 50 \text{ } \Omega; \qquad R_{max} = 150 \text{ } \Omega; \tag{125}$$

are used in these calculations.

for a Buck DC-DC converter without a PI controller ($L = 167$ µH; $C_{min} = 188$ nF).

In [20], conventional methods estimate the inductance L and the minimal capacitance $C_{min}$ in a similar Buck DC-DC converter in the following manner:

$$L = \frac{U_{Target} \left(E - U_{Target}\right)}{\Delta i_L F_s E}, \qquad C_{min} = \frac{\Delta i_L}{8 F_s \Delta u_R}, \qquad \Delta i_L = (0.2 \div 0.4) i_R \tag{126}$$

In case of $\Delta u_R = 0.02$ V, these formulas give the following results:

$$L = 125 \div 250 \text{ µH}; \qquad C_{min} = 125 \div 250 \text{ nF}; \tag{127}$$

which significantly differ from the values given in Figure 3c. Indeed, $K_p$ and $K_i$ have not been taken into account here (there is not a PI controller in the Buck DC-DC converter presented in [20]). Output voltage simulation results for the scheme, which is considered in the present paper, are shown on Figure 7. They are based on (126) and (127) with $\Delta i_L = 0.3 i_R$ and $C = 5 C_{min}$; Algorithm 3 with fixed values of L and C has been performed in the calculation of optimal values of the coefficients for the proportional and the integral terms in the PI controller.

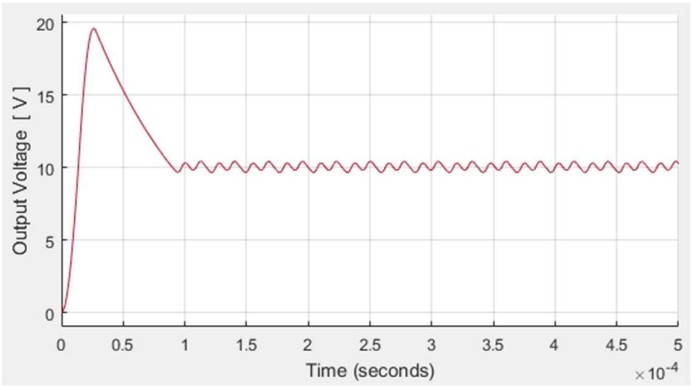

**Figure 7.** Output voltage based on an inductance and a minimal capacitance both determined.

The main advantage of PI controllers is that they can be implemented without detailed knowledge of systems. Algorithms of tuning PI controllers are presented in [21]. They consider devices that are regulated as black boxes. Therefore, they cannot change parameters L and C. Generally, estimations of $K_p$ and $K_i$ are empirically based on the simulation of a large number of simulations in these conventional algorithms. There exist well-established approaches to derive initial coefficients. The Ziegler–Nichols method is used below in order for a PI controller to be tuned [21,22]. The parameters obtained in Algorithm 3 are used (see Figure 3c):

$$L = 1.1 \text{ µH}; \qquad C = 450 \text{ µF}. \tag{128}$$

A proportional (P) controller with coefficient $K_p = 32$ is modeled at first; the value of P starts from 0 and increases until the system starts to show consistent and stable oscillation ($\Delta u_R \approx 0.01$ V). The next proportional controller is modeled with $K_p = 0.45 \times 32 = 14.4$. The final controller is proportional–integral with coefficients $K_p = 14.4$ and $K_i = 172.8$, which are close to these ones in Figure 3c. The results are presented in Figure 8.

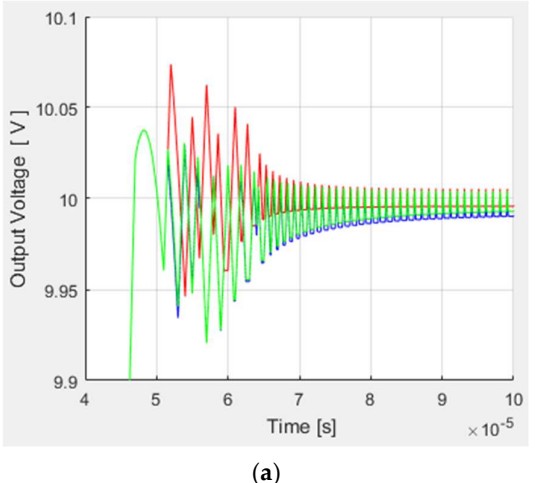

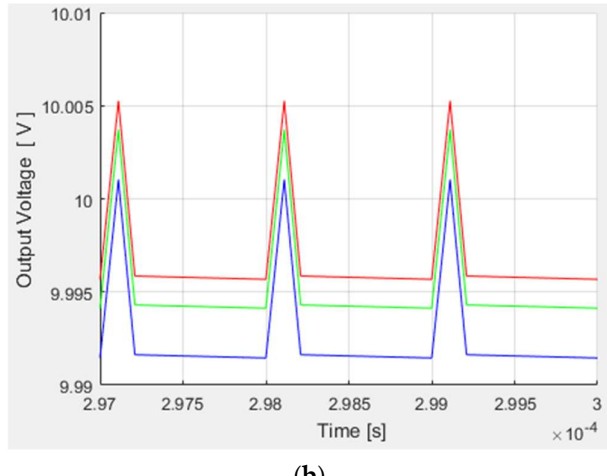

(**a**)                                                                                      (**b**)

**Figure 8.** Tuning PI controller according to Ziegler–Nichols method. Graphics with: $K_p = 32$, $K_i = 0$ (red one); $K_p = 14.4$, $K_i = 0$ (blue one); $K_p = 14.4$, $K_i = 172.8$ (green one); (**a**) maximal values; (**b**) oscillation at time 0.3 ms.

## 6. Simulation Duration

Three software applications modeling the discussed power converter are as follows: the Simulink model from Figure 1 and two index matrix-based models: the .NET one (which is described in [1]) and MATLAB one (which implements the algorithms from the previous two sections using MATLAB source code without using Simulink). Their runtimes are shown in Table 1. There is a slight difference in the .NET model: ideal switches have been used; all programming methods defined in the .NET software are presented in [1]. A single simulation of the converter operation is taken into account when calculating the durations. Each runtime considers both the time to run the simulation and the time to configure the model. For example, the Simulink model runtime is formed as a sum of execution times of two functions: set_params() and sim().

**Table 1.** Runtimes of software applications modeling Buck DC-DC converter.

| Model Time Duration (ms) | 0.1 | 0.2 | 0.3 | 0.4 | 0.5 |
|---|---|---|---|---|---|
| Runtime in Simulink (ms) | 622 | 728 | 848 | 943 | 1042 |
| Runtime in the authors' .NET software (ms) | 18.6 | 23.3 | 27.7 | 33.1 | 37.5 |
| Runtime in the authors' MATLAB software (ms) | 2.7 | 6.4 | 9.8 | 12.8 | 16.2 |

Runtimes for Simulink and .NET models determine the following linear regressions:

$$t_{run}^{Simulink} \approx 520 + 1055 t_{model\ time}; \tag{129}$$

$$t_{run}^{.NET} \approx 14 + 48 t_{model\ time}; \tag{130}$$

where all durations are measured in milliseconds. These expressions indicate the presence of a significant configuration time in these two models, especially for the first one. Here and below, $t_{model\ time}$ denotes a duration of model time. For the third model:

$$t_{run}^{MATLAB} \approx 33 t_{model\ time} \tag{131}$$

i.e., the runtime is proportional to the model time duration (this fact is conditioned by the pre-calculation of matrices—see Sections 4.1 and 4.2). There are commensurate coefficients (48 and 33 approximately) in the .NET and MATLAB models.

## 7. Conclusions

The proposed new definition of an R-IM product guarantees convenient index matrix equations with zero R-IM on the right-hand side. Index sets can determine which analog values should be calculated at any time in the model; additional values do not affect the calculation procedure.

The simulations produce results that are virtually identical to those from Simulink. At the same time, large changes in the ratio between the sampling rate and switching frequency keep the output characteristics almost unchanged. Threshold voltages (potential barriers) and dynamic resistances have been successfully included in the models.

The performed optimization procedures based on exhaustive searches with nested intervals give very good results—small peaks and small deviations after settling. It is shown that when the choice of inductance and capacity is made, the tuning of the PID controller by means of the proposed algorithm is close to that which has been achieved with conventional methods.

The discussed MATLAB implementation based on the pre-computation of indexed matrices and the selection of variants results in small computation times.

**Author Contributions:** N.H., P.G. and V.G. were involved in the full process of producing this paper, including conceptualization, methodology, modeling, validation, visualization and preparing the manuscript. All authors have read and agreed to the published version of the manuscript.

**Funding:** This research was funded by Bulgarian National Scientific Fund, grant number КП-06-Н57/7/16.11.2021, and the APC was funded by КП-06-Н57/7/16.11.2021.

**Data Availability Statement:** Data are contained within the article.

**Acknowledgments:** This research was carried out within the framework of the project "Artificial Intelligence-Based modeling, design, control and operation of power electronic devices and systems", КП-06-Н57/7/16.11.2021, Bulgarian National Scientific Fund.

**Conflicts of Interest:** The authors declare no conflict of interest.

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
