# Peer review of "Index Matrix-Based Modeling and Simulation of Buck Converter"

_mathematics, doi:10.3390/math11234756_

Round 1

Reviewer 1 Report

Comments and Suggestions for Authors

Comments of this reviewer on the manuscript Mathematics-2705880 are as follows:

1.     The topic of this manuscript belongs to the scope of the journal and according to the best knowledge of this reviewer, it can be considered novel.

2.     It is obvious that the content of this manuscript offers a sufficient amount of information necessary to repeat the presented results.

3.     The number of equations in this manuscript is large, but omitting a certain number of these equations would jeopardize the completeness of the content.

4.     The obtained results confirm that the proposed model is well developed. In particular, a good agreement was obtained between the results generated by different software tools.

5.     The only thing that needs fixing in this manuscript is the language. There are some phrases and some errors that must be corrected. For example: “The described in thе paper an approach…”, “…simple stability analisis…”, “…similar to the presented in…”, “In the considered…”, “…which is shown on Fig. 1,…”, “These statements show that upper index matrices, which identifiers…”, “…are shown on Fig. 2.”, etc.

6.     Keywords should be in singular form and listed in alphabetical order.

7.     In the introduction, there are lumped citations such as “…quite different areas [4 - 9].”, “…and inputs [5 - 8].”, and “…(such as [17 - 20])…”.  Literature cannot be reviewed in this way, and the lumped citations should be split into normal citations.

Comments on the Quality of English Language

Minor editing of English language required

Author Response

First of all, I would like to thank you for your thorough review of our paper (mathematics-2705880) and helpful comments to improve it.

To Reviewer 1:

            Thank you very much for your review and valuable remarks.

 The topic of this manuscript belongs to the scope of the journal and according to the best knowledge of this reviewer, it can be considered novel.

- Thank you very much for the high rating of our work. The main idea of the manuscript is to present a new apparatus for modeling DC-DC converters, through which to achieve optimal design of both the power circuit and the adjustment of the controller coefficients.

  1. It is obvious that the content of this manuscript offers a sufficient amount of information necessary to repeat the presented results.

- Thank you very much for your comment. Our idea is to present enough information to allow the manuscript to be used independently, rather than refer the reader to multiple relevant sources.. 

  1. The number of equations in this manuscript is large, but omitting a certain number of these equations would jeopardize the completeness of the content.

- Thank you very much for your comment. The equations are necessary for a complete description of the method.

  1. The obtained results confirm that the proposed model is well developed. In particular, a good agreement was obtained between the results generated by different software tools.

- Thank you very much for the rating! Achieving a good match between the results generated by different software tools was one of our main goals.

  1. The only thing that needs fixing in this manuscript is the language. There are some phrases and some errors that must be corrected. For example: “The described in thе paper an approach…”, “…simple stability analisis…”, “…similar to the presented in…”, “In the considered…”, “…which is shown on Fig. 1,…”, “These statements show that upper index matrices, which identifiers…”, “…are shown on Fig. 2.”, etc.

- Thank you very much for the remark. The manuscript has been revised and edited.

  1. Keywords should be in singular form and listed in alphabetical order.

- Thank you very much for the remark. Corresponding corrections have been made.

  1. In the introduction, there are lumped citations such as “…quite different areas [4 - 9].”, “…and inputs [5 - 8].”, and “…(such as [17 - 20])…”. Literature cannot be reviewed in this way, and the lumped citations should be split into normal citations.

- Thank you very much for the remark. Corresponding corrections have been made.

Comments on the Quality of English Language

Minor editing of English language required

The manuscript has been substantially reviewed and edited.

Thank you very much for your remarks and comments. They were very useful for me to emphasize the main tasks and contributions of the manuscript, and also to focus the attention of the readers on the new and unique elements!

 Reviewer 2

Comments to the Authors

My comments

The method for managing the parameters and features of electronic circuits through the use of index matrices is presented in this research. These smaller, more descriptive models give precise explanations of operations. Because of switching topologies and component shortages, the authors contend that they are still relevant in power electronics. They give an example of a power converter model and show how to use this strategy using MATLAB source code.

The paper is well written, especially for students and researchers. I recommend accepting the paper.

To Reviewer 2:

            Thank you for your review and valuable remarks.

  1. The method for managing the parameters and features of electronic circuits through the use of index matrices is presented in this research. These smaller, more descriptive models give precise explanations of operations. Because of switching topologies and component shortages, the authors contend that they are still relevant in power electronics. They give an example of a power converter model and show how to use this strategy using MATLAB source code.

The paper is well written, especially for students and researchers. I recommend accepting the paper.

- Thank you very much for the high rating of our work! The main idea of the manuscript is to present a new apparatus for modeling DC-DC converters, through which to achieve optimal design of both the power circuit and the adjustment of the controller coefficients. We hope that our research will both contribute to the improvement of power electronics education and be useful to designers of power electronic devices and systems.

Reviewer 3 Report

Comments and Suggestions for Authors

Review Report: mathematics-2705880-peer-review-R1

Title: the title is inappropriate based on the contents of the manuscript.

Suggestion: Index Matrix Based Modeling and Simulation of Bulk Converter

Abstract: I have the following observations about the abstract:

i.                     The abstract should re-written to clearly present the problem, method, and results.

ii.                   There are a few typos.

Introduction: The following comments are to be addressed on the introduction:

i.                     There should a section dedicated to notations and symbols

ii.                   This sentence in Lines 54-56 is vague. This refers to what?

iii.                 There are instances of poor use of English language, typos, and missing punctuation.

iv.                  The introduction can be improved upon.

Literature Review: N/A

Methodology: The methodology is highly chaotic considering the mathematical nature of this work. The authors should try to demystify the method by presenting the most relevant equations only and subsequently explain them.  Also, the following should be taken care of:

i.                     This should be L. except capacitor with capacitance C is intended (Line 175)

ii.                   Please indicate which of the equation 38-40 represent target voltage, PI controller, and PWM

iii.                 What does this “Statement (40)” mean? It is quite confusing (Line 219)

iv.                  The sentence in Line 227 – 228 is ambiguous. Please recast.

v.             Which equation describe [S]? (Line 230)

vi.                  How are the values in equation (51) gotten? (Line 241)

vii.                What are B'(t) and J'(t) in Line 294?

viii.              The use of "on" in place of "of" is prevalent in this manuscript.

ix.                  What is the full meaning of the parameters listed in Line 407 – 410?

Results: The results were not clearly presented and should be reworked.

i.                     Figure 2 – 6 are poorly explained. Please address this.

ii.                   How was the value in equation (105) determined or was it just a selection? (Line 485)

iii.                 Section 5 presented Optimization however, the optimized results were not clearly presented. There should be a comparison of the optimized results with the conventional results.

iv.                  No result is presented for the classical method. It is expected that classical results would be presented also to enable comparative analysis.

v.                   Table 1 is poorly explained. Please address this.

Conclusion:

The conclusion should be improved upon by itemizing the highlights of this study. Some of the claims in this section lack adequate results to justify them.

Reference:

Citation of references 22 – 24 cannot be found in the body of the work.

Plagiarism:

A plagiarism of 11% was detected using Turnitin software. This is considered to be fairly okay.

Comments on the Quality of English Language

The use of need to be improved upon to enhance readability of this work.

Author Response

First of all, I would like to thank you for your thorough review of our paper (mathematics-2705880) and helpful comments to improve it.

To Reviewer 3:

            Thank you for your review and valuable remarks.

In the attached file you will find detailed point-by-point responses to your remarks, comments and recommendations.

Thank you very much for your remarks and comments. They were very useful for me to emphasize the main tasks and contributions of the manuscript, and also to focus the attention of the readers on the new and unique elements.

 Again thank you all for the exact review!
